# Tips for Preparing and Practicing Thermal Ablation Therapy of Hepatocellular Carcinoma

**DOI:** 10.3390/cancers15194763

**Published:** 2023-09-28

**Authors:** Yasunori Minami, Tomoko Aoki, Satoru Hagiwara, Masatoshi Kudo

**Affiliations:** Department of Gastroenterology and Hepatology, Faculty of Medicine, Kindai University, 377-2 Ohno-Higashi Osaka-Sayama, Osaka 589-8511, Japanm-kudo@med.kindai.ac.jp (M.K.)

**Keywords:** percutaneous thermal ablation, hepatocellular carcinoma, image guidance, radiofrequency ablation, microwave ablation

## Abstract

**Simple Summary:**

Thermal ablation, including radiofrequency ablation (RFA) and microwave ablation (MWA), for hepatocellular carcinomas (HCCs) is accepted as a curative treatment option in many HCC treatment guidelines because of the better clinical outcomes performed in a more minimally invasive manner. Essential management tips to successful ablation therapy can be organized into five categories: understanding the principles of ablation therapies and the characteristics of ablation devices, assessing the benefits and risks of ablation, practicing ultrasound-guided needle tip control and visualization, taking advantage of imaging guidance techniques, and evaluating therapeutic response to ensure adequate ablation. We herein provide an overview of the basic principles and characteristics of tissue heating, highlight the safety management of ablation therapy, and provide the technical skills of sophisticated planning and image guidance technologies to improve short- and long-term outcome of HCC patients. These essentials will contribute to the expansion of thermal ablation applications in clinical settings.

**Abstract:**

Thermal ablation therapy, including radiofrequency ablation (RFA) and microwave ablation (MWA), is considered the optimal locoregional treatment for unresectable early-stage hepatocellular carcinomas (HCCs). Percutaneous image-guided ablation is a minimally invasive treatment that is being increasingly performed because it achieves good clinical outcomes with a lower risk of complications. However, the physics and principles of RFA and MWA markedly differ. Although percutaneous thermal ablation under image guidance may be challenging in HCC cases with limited access or a risk of thermal injury, a number of ablative techniques, each of which may be advantageous and disadvantageous for individual cases, are available. Furthermore, even when a HCC is eligible for ablation based on tumor selection and technical factors, additional patient factors may have an impact on whether it is the appropriate treatment choice. Therefore, a basic understanding of the advantages and limitations of each ablation device and imaging guidance technique, respectively, is important. We herein provide an overview of the basic principles of tissue heating in thermal ablation, clinical and laboratory parameters for ablation therapy, preprocedural management, imaging assessments of responses, and early adverse events. We also discuss associated challenges and how they may be overcome using optimized imaging techniques.

## 1. Introduction

Locoregional therapies are defined as minimally invasive image-guided liver tumor-directed procedures that can be categorized into ablative therapy, transcatheter therapy, and radiation therapy. Ablative therapies for hepatocellular carcinomas (HCCs) are classified as either chemical ablation (including ethanol or acetic acid injection) and thermal ablation (including laser, radiofrequency, microwave, cryoablation, and high-intensity focused ultrasound). Percutaneous ethanol injection (PEI) was the first ablative technique used for treating early-stage hepatocellular carcinomas (HCCs), where absolute alcohol is injected into the tumor. However, PEI therapy has been replaced by newer and more effective thermal ablation techniques due to its high local recurrence rate. 

Thermal tumor ablation therapy can destroy an entire tumor by using heat to kill malignant cells and is one of the promising minimally invasive techniques for the treatment of nonresectable HCCs. Thermal ablation, including radiofrequency ablation (RFA) and microwave ablation (MWA), offers the advantages of being less invasive due to the use of a percutaneous procedure with ultrasonography (US) or computed tomography (CT) as the guiding modality and the lower risk of major complications. Therefore, recurrent small HCCs are also eligible for repeat ablation. Today, percutaneous RFA and MWA are considered the standard local ablative modalities for the treatment of early-stage HCCs. From a technical standpoint, complete, accurate, and safe ablation is essential to achieve the best outcomes in ablation for HCCs. Many technological advances are continually introduced to improve upon the effectiveness of thermal ablation. To tailor therapy to a specific patient’s condition (e.g., thrombocytopenia, coagulopathy, ascites, history of biliary surgery or chronic renal disease, etc.), careful monitoring can help prevent complications after ablation. 

Essential management tips to successful ablation therapy can be organized into five categories: understanding the principles of ablation therapies and the characteristics of ablation devices, assessing the benefits and risks of ablation, practicing ultrasound-guided needle tip control and visualization, taking advantage of imaging guidance techniques, and evaluating therapeutic response to ensure adequate ablation. Due to the rising complexity of treatment devices and expanding paradigms for tumor ablation, an understanding of the basic principles of ablation is a prerequisite for the effective treatment of HCCs. We herein highlight various aspects of and safety management by thermal ablation therapy. We also discuss thermal ablation techniques in recent imaging and technological guidance. These essential tips will be very helpful for physicians performing successful ablations of HCCs.

## 2. Principles of Thermal Ablation and Treatment Devices

### 2.1. Radiofrequency Ablation (RFA)

RFA is a current loop comprising a generator, cabling, electrodes, and the grounding pad (Figure 1). An electrical current of ~450–500 kHz causes the ions inside the tissue to rapidly oscillate, resulting in resistive tissue heating around the active electrode, which is known as the Joule effect [1]. This induces local temperatures of between 60 and 100 °C (Table 1).

A 17 gauge electrode needle-type probe is employed and connected to a 200-watt RF generator. There are two main types of electrodes for RFA: multiple expandable electrodes (such as StarBurst^®^ (Rita Medical Systems; Mountain View, CA) and LeVeen^®^ (Boston Scientific; Natick, MA, USA)) and internally cooled electrodes (including Cool-tip^®^ (Covidien; Mansfield, MA, USA)). A bipolar RFA system (CelonPOWER^®^ (CelonLabPower; Teltow, Germany)) has been established with multiple cooled electrodes in the absence of a grounding pad. A RFA electrode was recently developed to adjust the ablation range. Using a sliding insulation sheath, the active metallic tip of the RFA needle (such as VIVA^®^ (STARMed; Goyang, Gyeonggi-do, Republic of Korea), Vari Tip^®^ (BVM Medical; Hinckley, UK), and arfa^®^ (Japan Lifeline; Tokyo, Japan)) may be adjusted in 5 mm intervals up to a length of 4 cm. Therefore, the length of the active tip may be selected according to the size, location, and shape of the tumor.

### 2.2. Microwave Ablation (MWA)

Electromagnetic energy operating at 915 MHz or 2.45 GHz may be used to produce very high (>150 °C) temperatures, which is known as an induction heating [1]. Microwave heating is caused by the rotation of water molecules, and the process differs from the Joule heating mechanism that underpins RFA (Figure 2). In addition, MWA does not require a grounding pad because an electric current does not flow through a patient’s body (Table 1). 

Recent developments and improvements in technologies have created a new microwave ablation system (Emprint^®^ (Covidien/Medtronic; Minneapolis, MN, USA) and MIMA Pro^®^ (Mima-pro Scientific Inc.; Nantong, China)). The Emprint ^®^ antenna probe is only 13 G, whereas Mimapro ^®^ has three types: 14 G, 17 G, and 18 G. An electrode needle-type probe is connected to a 100-watt MW generator. The circulation of sterile saline solution down the shaft to the distal probe tip internally cools the probe antenna and cables, which achieves thermal control and provides a reliable and stable ablation zone near the antenna shaft during tissue desiccation. The desired field shape is obtained by microwave field control, despite changes in the tissue environment by ablation. Furthermore, heat-induced changes in the dielectric constant immediately around the probe antenna are minimized by wavelength control [2]. Therefore, larger ablation zones are generated by MWA. The ablation time is shorter, the ablation zone is larger, and the heat sink effect is weaker with MWA than with RFA. In clinical settings, MWA and RFA are considered to have equivalent efficacy, complication rates, local recurrence rates, and survival rates [3,4].

### 2.3. Potential Immunomodulatory Effects

Thermal ablation involves the induction of irreversible damage to cancer cells by localized heat and may result in the release of tumor antigens. Thermal ablation-induced inflammation and increases in tumor antigens can be expected to promote the cancer immunity cycle. The combination of immunotherapy and thermal ablation may be an emerging therapeutic option with enhanced efficacy [5,6,7].

## 3. Clinical Indications and Proper Methodology

Percutaneous thermal ablation is commonly performed on patients with HCCs in Eastern and Western countries, generally for Child–Pugh class A or B with ≤3 tumors of a diameter ≤3 cm [8,9,10,11,12]. According to international HCC clinical practice guidelines, there is no difference in treatment indication between RFA and MWA. We should understand the ablation mechanisms and properties of therapeutic needles for maximizing therapeutic effects. Intermediate-stage HCCs or lesions larger than 5 cm might also be treated with ablation if effective local tumor control is achieved with available ablation and guidance techniques. Moreover, percutaneous thermal ablation can also be effective and safe for elderly patients, and its clinical benefits do not appear to be negatively affected by comorbidities [13,14]. 

However, relatively large tumors (>2.5–3 cm) have been identified as a predictor of local recurrence, but not overall survival, after RFA [15]. Therefore, RFA needs to be selected according to not only technical feasibility and tumor sizes, but also the severity of portal hypertension if present, and the remaining liver volume and expected liver function after surgery in each patient.

## 4. Tips for Beginning Tumor Ablation

### 4.1. Planning US and Software-Based Planning

A simulation of the ablation procedure allows operators to treat patients more efficiently, which results in shorter procedure times and better outcomes. Planning US is generally performed one day before treatment to establish whether percutaneous ablation therapy is feasible. It has the advantage of planning ablation strategies, including the therapeutic needle path and placement, and overlapping ablations [16,17]. 

Software-assisted planning and simulations are also beneficial for percutaneous ablation. Graphics software (including Advantage Workstation^®^ (GE Healthcare; Waukesha, WI, USA), SYNAPSE VINCENT^®^ (Fujifilm Medical; Tokyo, Japan), Ziostation2^®^ (Ziosoft Inc.; Tokyo, Japan), AZE Virtual Place^®^ (Canon Medical; Tokyo, Japan), and BioTrace IO^®^ (TechsoMed; Rehovot, Israel)) display the following scenarios: a pre-interventional simulation, including the orientation and positioning of a tumor and the optimization of access paths, and a peri-interventional simulation, including predictions of the ablation zone and thermal organ injury [18,19].

### 4.2. Grounding Pad on the Back

Temperatures at the grounding pad are dependent on a number of variables, including the grounding pad surface area, the amount of current deposited in the liver, the orientation of the pad, and the pad’s distance from the RF electrode [20]. For example, a decrease in the distance between the electrode and the pad facilitates the flow of electricity. Therefore, lower impedance between the RF electrode and the ground pad means that current density (and, thus, heating) around the RF electrode is increased. Although the grounding pad is generally fixed to a patient’s thighs, placement on the back is favorable for shortening the ablation time without reducing the ablation zone and increasing pain [21].

### 4.3. Body Position Change

The liver may be shifted by a patient’s body position, which may help identify the tumor site when sonogram poorly visualizes the liver. Patients are generally placed in the supine position for a percutaneous approach. The semi-Fowler position may be useful for showing a HCC in the right hepatic lobe, particularly the subphrenic area, due to descent down the liver [22], while left lateral decubitus positioning may allow for imaging over the liver because the bowels, which contain intraluminal air, are less likely to be present in right subcostal scanning [23].

### 4.4. Control of Needle Insertion on Clear US Images

The precise positioning of needle insertion through the center of a tumor is an essential task for complete tumor ablation. However, the needle tip is sometimes difficult to discern under the guidance of US, and a failure to visualize the needle tip during needle advancement may result in a technical error. The difficulty associated with aligning the needle and US transducer is one of the most common factors contributing to inadequate needle visualization [24]. Therefore, advancing the needle without observing the tip properly must be avoided. Cooperative manipulation between the US probe and the therapeutic needle is challenging because of the dynamic nature of US and, thus, requires extensive practice.

### 4.5. Extreme Steam Popping

Extreme steam popping may cause subcapsular or intraperitoneal hemorrhage during ablation for HCCs [25,26]. Low-power ablation may delay steam popping [27]; therefore, the low-power technique of the RFA protocol with manual multistep increments in power is widely used to prevent a rapid increase in intertumoral pressure. New RFA systems (such as the VIVA RF system^®^ (STARMed; Goyang, Gyeonggi-do, Republic of Korea) and the arfa RF ablation system^®^ (Japan Lifeline; Tokyo, Japan)) have recently been equipped with the linear mode of ablation, with gradual increases of 5 or 10 watts each minute automatically until the break of energization. Additionally, impedance monitoring is extremely useful for predicting the occurrence of popping in RFA. A temporary power outage may help prevent extreme steam popping when a hyperechoic zone cannot be visualized around the needle despite a higher voltage for ablation.

## 5. Risk Assessment for Tumor Ablation

### 5.1. Low Platelet Count (Thrombocytopenia)

Although hemorrhagic complications are rare with percutaneous ablation therapy, thrombocytopenia or platelet dysfunction in patients with chronic liver disease may increase the risk of post-operative bleeding following invasive procedures [28], and platelet transfusions may be advised in clinical practice for platelet counts <50,000/μL. Additionally, lusutrombopag, an orally active, small-molecule thrombopoietin receptor agonist that induces the production of endogenous platelets, may be used as prophylaxis to reduce the risk of bleeding [29,30]. However, the use of platelet transfusions to prevent post-operative bleeding is controversial because it has not yet been established whether prophylactic platelet transfusion is superior to no prophylaxis, if the platelet count threshold is appropriate as a guide to initiate the transfusion of prophylactic platelets, and what dose of platelets is needed to prevent post-operative bleeding [31,32].

### 5.2. Low Levels of Blood Clotting Factors (Coagulopathy)

Routine coagulation screening is typically performed to predict post-operative bleeding in patients with liver cirrhosis. Before an invasive intervention, fresh frozen plasma (FFP) transfusion is considered for patients with a prothrombin time–international normalized ratio (PT-INR) ≥ 1.5; however, the thresholds used for PT-INR correction in clinical settings lack supportive evidence. The routine correction of thrombocytopenia and coagulopathy in patients with liver dysfunction-induced coagulation abnormalities prior to a low-risk ablation procedure is generally not recommended [33]. Furthermore, a previous study demonstrated that the use of FFP transfusion to reduce PT-INR and expedite interventions was ineffective in critically ill patients with coagulopathy associated with liver cirrhosis [34].

### 5.3. Ascites, Esophagogastric Varices, and Hepatic Encephalopathy

HCC patients with poor liver function may develop ascites, variceal hemorrhage, or hepatic encephalopathy. The treatment of these symptoms needs to be initiated immediately prior to an invasive intervention for HCCs, and locoregional therapies are generally planned after the resolution of these symptoms. However, some HCC patients with persistent symptoms may receive locoregional therapies. For example, a percutaneous procedure including ablation and biopsy is widely considered a contraindication in the presence of ascites due to the risk of uncontrollable bleeding into ascites [35]. Previous studies reported that image-guided liver interventions in the presence of ascites did not affect the post-operative hemorrhage rate [36,37]. Salvage locoregional therapy may prolong overall survival, even in patients with a higher Child–Pugh score [38]; however, the decision regarding eligibility for ablation requires careful consideration.

### 5.4. History of Biliary Surgery/Interventions

Post-ablation infections, including cholangitis or liver abscess, are rare, and there is no consensus on the effectiveness of prophylactic antibiotics for patients undergoing thermal ablation to reduce the risk of postprocedural infection [39,40]. However, the high risk of cholangitis in patients who underwent biliary surgery or endoscopic sphincterotomy is primarily caused by failed or incomplete biliary drainage [41,42], and bilioenteric anastomosis strongly correlated with the development of cholangitis, potentially leading to severe complications. Therefore, patients with a history of biliary surgery or endoscopic sphincterotomy require more rigorous antimicrobial prophylaxis.

### 5.5. Comorbidity of Renal Failure

Difficulties are associated with the treatment of HCCs in patients with chronic kidney disease (CKD) because CKD itself is associated with a 1.5-fold increased risk of bleeding [43]. Even if HCC patients complicated with renal failure have a normal platelet count, increased capillary fragility and the disturbance of blood coagulation may result in post-operative hemorrhage. Therefore, close attention to severe CKD is needed to prevent post-operative bleeding [44,45].

## 6. Management of Factors Affecting the Visualization of Tumors in US

Achieving a complete pathologic response by ablation for HCCs requires a transient ablative hyperechoic zone to cover it entirely during a procedure. In short, the needle is introduced percutaneously into the center of the tumor nodule on B-mode US, and the needle tip is positioned at the deepest margin of the target tumor. Multiple overlapping ablation technique is sometimes required due to the geometry of the tumor. However, it can be challenging to ablate HCCs under B-mode US guidance when HCCs have poor sonographic conspicuity. Then, we need some progressive approaches to cancer detection. 

### 6.1. Artificial Ascites/Pleural Effusion Technique

The creation of artificial ascites or pleural effusion is a supportive ablation technique for HCCs located close to the gastrointestinal tract or diaphragm [46,47,48]. Percutaneous image-guided ablation for HCCs in these high-risk locations is technically challenging due to the risk of thermal injury to the surrounding organ and/or the poor sonographic visualization of a tumor caused by an overlap with the intestines or lungs. Intraperitoneal or intrapleural infusion may act as an acoustic window and a thermal blanket by ensuring the adequate separation of the ablation zone from the adjacent organ.

### 6.2. Contrast-Enhanced US (CEUS) Guidance

Microbubble contrast agents are widely used in US imaging. Sulfur hexafluoride microbubbles (SonoVue^®^; Bracco SpA, Milan, Italy), perflutren lipid microbubbles (Definity^®^; Bristol-Myers Squibb, North Billerica, MA), perflutren protein microbubbles (Optison^®^; GE Healthcare, Buckinghamshire, UK), and perfluorocarbon microbubbles (Sonazoid^®^; GE Healthcare, Oslo, Norway) are second-generation contrast agents. These microbubbles provide stable nonlinear oscillation in a low power acoustic field because of their hard shells, producing great detail in the harmonic signals in real time. Dynamic CEUS displays similar, but distinct, vascular patterns to dynamic CECT; the US contrast agents are retained within blood vessels (blood pool contrast agents), whereas those for CT and magnetic resonance imaging (MRI) move into the extracellular space until their concentrations balance between the intravascular and extracellular spaces. Meanwhile, the Sonazoid^®^ can be taken up by Kupffer cells in the liver, and Sonazoid^®^ microbubbles accumulate in the liver parenchyma over time (Figure 3).

CEUS increases conspicuity and more accurately characterizes hypervascular HCCs that are poorly visualized in B-mode US. HCCs are visualized as defects in the liver parenchyma during the Kupffer phase, only with Sonazoid use. Therefore, these defect lesions can be used as a target for the insertion of a single needle. Meanwhile, advanced skill is required with SonoVue^®^, Definity^®^, or Optison^®^ use in CEUS guidance because the optimum timing is too short to search for enhanced HCC nodules and insert the therapeutic needle during the early vascular phase (within approximately two minutes after injection). 

CEUS guidance in ablation can improve diagnostic performance through the correct targeting of HCCs and real-time needle navigation with good short-term treatment responses. The technical success rate of a single RFA session was significantly higher with CEUS than with B-mode US (94.7% vs. 65.0%, *p* = 0.043) [49]. Furthermore, the number of RFA sessions conducted in a historical cohort was smaller with Sonazoid CEUS guidance than with B-mode US guidance [50]. Another study showed that the sustained local control rate was markedly higher for CEUS-guided RFA than for B-mode US-guided RFA (85.3% vs. 66.4% at 2 years) [51]. 

### 6.3. Fusion Imaging Guidance

Advances in technology have led to the introduction of imaging techniques that combine CT or MRI and US in clinical practice, and fusion imaging has emerged as a valuable guide for ablating small HCCs with poor conspicuity in US [52,53,54]. Image fusion has evolved into a relatively mature option for high-end US machines, with many manufacturers offering various options. The names of these US machines highlight the nature of their function: Real-time Virtual Sonography^®^ (Fujifilm Medical; Tokyo, Japan), Volume Navigation^®^ (GE Healthcare; Wauwatosa, WI, USA), PercuNav^®^ (Philips Healthcare; Bothell, WA, USA), Smart Fusion^®^ (Canon Medical; Tokyo, Japan), and eSie Fusion Imaging^®^ (Siemens Healthcare; Forchheim, Germany). The movement of the US transducer allows for the display of two-dimensional (2D) multiplanar reconstruction (MPR) images from CT or MRI in the same plane as US images (Figure 4). Thus, CT/MR–US fusion imaging can improve the visualization of inconspicuous HCCs and helps us to understand the three-dimensional relationship between the liver vasculature and HCCs. Furthermore, the operator confidence is increased by targeting with CT/MR-US fusion imaging techniques, and the technical success rates for HCCs with poor conspicuity ranged from 94.4 to 100%. 

Treatment responses may also be monitored during ablation using US fusion imaging. Moreover, fusion imaging allows for side-by-side comparisons of real-time 2D (post-ablation) and MPR (pre-ablation) US images as well as easy visualization of the ablative margin during ablation. US–US fusion imaging allows side-by-side comparison of the ablative margin during the ablation, and this feedback helps operators to recognize residual tumors. Moreover, US–US overlay fusion can visualize the ablative margin immediately after ablation because the tumor image is projected onto the ablative hyperechoic zone. Previous studies demonstrated that a sufficient ablative margin was achieved using US-US overlay fusion guidance, which ultimately reduced the risk of local tumor progression [55,56]. 

A flickering screen in image fusion may occur when the US probe with a magnetic sensor becomes distant from the magnetic generator. The magnetic field roughly extends up to 60 cm ahead of the magnetic generator. Therefore, the generator should be set up above the patient’s body; keeping a close distance between the generator and the US probe is needed for fusion imaging. 

## 7. Treatment Response Assessment

Ablative heating leads to tissue dehydration and water vaporization, and a transient hyperechoic ablated zone arises due to the generation of vapor bubbles as strong acoustic scatterers. Therefore, the HCC needs to be fully covered by an ablative hyperechoic zone, regarded as a necrotic lesion, during the ablation procedure [57,58]. 

Multiphasic CT or MRI with a contrast material is routinely used for post-ablation surveillance imaging [59,60,61,62,63], and the first follow-up imaging is usually scheduled four to six weeks after therapy. In the follow-up period, the patient may develop not only local tumor progression but also new intrahepatic recurrence, and clinical long-term follow-up will provide support to successfully detect these lesions. The aims of post-ablation imaging are three-fold: 1, to assess the technical success of the ablation and identify immediate adverse events; 2, to accurately detect tumor progression and intrahepatic recurrences; and 3, to identify any distant extrahepatic recurrences. Ablation is deemed successful when both the absence of tumor vascular enhancement and a sufficient ablative margin are achieved. The safety margin of ablation is regarded to be >5 mm to avoid the risk of local tumor progression, because microsatellite lesions may be distributed around the HCC nodule [64,65]. Therapeutic responses are assessed by comparing axial images before and after ablation, generally in a side-by-side manner. If the residual HCC is obscured in the reactive hyperemic region, side-by-side comparisons and measurements of the ablative margin may result in false and misleading ablation treatment assessments. Graphics interface software (such as Advantage Workstation^®^ (GE Healthcare; Waukesha, WI, USA), Syngo.via VB20A^®^ (Siemens Healthcare; Forchheim, Germany), SYNAPSE VINCENT^®^ (Fujifilm Medical; Tokyo, Japan), Ziostation2^®^ (Ziosoft Inc.; Tokyo, Japan), AZE Virtual Place^®^ (Canon Medical; Tokyo, Japan), and BioTrace IO^®^ (TechsoMed; Rehovot, Israel)) allows for the easy overlay of images obtained pre- and post-ablation therapy and supports the evaluation of HCC ablative margins three-dimensionally to overcome the potential bias in subjective evaluations [66,67]. 

## 8. Clinical Outcomes and Adverse Events

Several studies show thermal ablation therapies, including RFA and MWA, to be as effective as surgical resection in HCCs ≤ 3 cm, with a 5-year overall survival rate of 60–80% [68,69,70] and a 2-year local recurrence rate of 1.7–24% [71]. The combination of precise ablation and accurate treatment response assessment using three-dimensional imaging techniques has promising potential to improve quality of cancer care and achieve better clinical outcomes. 

The overall rates of early mortality after percutaneous ablation were previously reported to range between 0.1 and 0.6%, while those of serious adverse events were between 2.2 and 7.8% [72,73,74]. Common early complications included liver abscess (0.2–0.8%), hemothorax (0.2–0.8%), gastrointestinal perforation (0.3–0.5%), peritoneal hemorrhage (0.2–0.5%), hepatic infarction (0.2–0.5%), pneumothorax (<0.2%), portal vein thrombosis (<0.2%), pleural effusion requiring medication (<0.2%), ground pad burns (<0.2%), and biliary hemorrhage (<0.1%).

## 9. Clinical Implications of Thermal Ablation Therapies

An alternative approach for the prevention of HCC recurrence may be to find a clinically available compound that is inexpensive, easily manageable, and less toxic, with a proven safety profile on long-term administration.

Vitamin K plays a role in controlling cell growth, and vitamin K2 can induce the differentiation of human myeloid leukemia cells, as well as apoptosis in immature blast cells. If vitamin K2 could reduce HCC recurrence by preventing carcinogenesis or suppressing tumor growth, vitamin K2 would be an ideal adjuvant agent. However, the efficacy of vitamin K2 in suppressing HCC recurrence was not confirmed in this double-blind, randomized, and placebo-controlled study [75].

Sorafenib inhibits both mitogen-activated protein kinase/extracellular signal-regulated kinase (MAPK/ERK)-mediated cell proliferation and angiogenesis driven by vascular endothelial growth factor (VEGF) signaling; this provides a nearly 3-month median survival benefit and a 31% reduction of risk of death in patients with advanced HCCs [76]. Despite the promising results of some reports and the theoretical advantages of sorafenib in an adjuvant setting, a broad multicenter randomized controlled trial (Sorafenib as Adjuvant Treatment in the Prevention of Recurrence of Hepatocellular Carcinoma (STORM)) failed to find a significant improvement in progression-free survival (primary endpoint) and overall survival [77]. 

Angiotensin-converting enzyme (ACE) inhibitors are currently widely used as anti-hypertensive agents in clinical practice. In addition, ACE inhibitors may be an alternative anti-angiogenic strategy in the treatment of liver fibrosis and HCCs, because angiogenesis is an essential process in tumor growth and liver fibrosis. In a retrospective cohort, ACE inhibitors significantly improved overall survival and the time to recurrence after RFA in HCC patients, in comparison with both patients under an ACE inhibitor and those not receiving any of the drug classes [78]. However, further prospective research with larger samples is warranted.

## 10. Conclusions

Based on standards recommended in international guidelines, percutaneous thermal ablation for early-stage HCCs is a safe, feasible, and potentially curative treatment. Due to the enrichment of knowledge and technological advances in the field of percutaneous ablation, personalized approaches are proposed for patients with early-stage HCCs. The technical skills, hardware and software requirements, and combination of different treatment techniques needed to improve the short- and long-term outcomes of HCC patients require specialized interventional oncologic centers with sophisticated planning, image guidance, and image fusion techniques. Adequate training, a basic understanding of the underlying principles and an awareness of the working mechanisms, as well as knowledge of the advantages and disadvantages of the different ablation techniques available, are needed by operators of thermal ablation. A number of skills are critical for the success of this procedure, including accurate positioning of the needle applicator under image guidance. Furthermore, knowledge of the expected imaging characteristics of successful and failed ablation, in addition to perioperative complications, is important. Collectively, these factors will contribute to the expansion of thermal ablation applications in clinical settings.

## Figures and Tables

**Figure 1 cancers-15-04763-f001:**
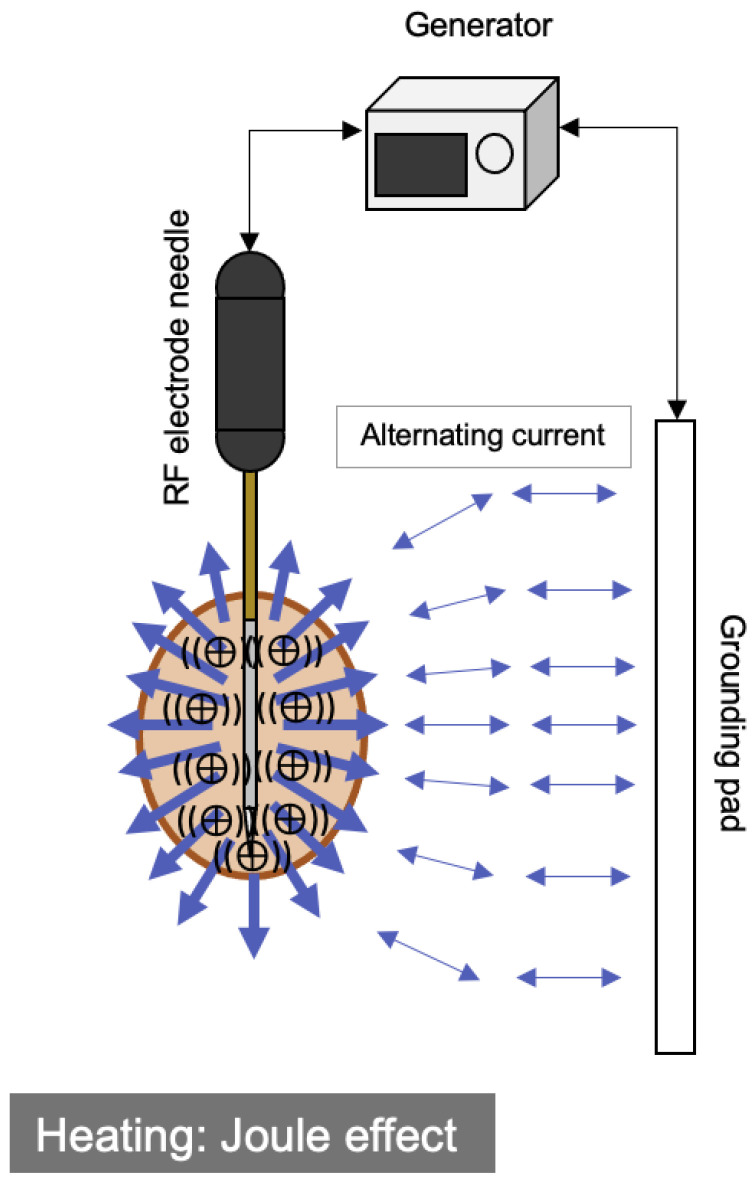
Schematic diagram of the RFA system and its mechanism of heat generation. Note, RF: radiofrequency.

**Figure 2 cancers-15-04763-f002:**
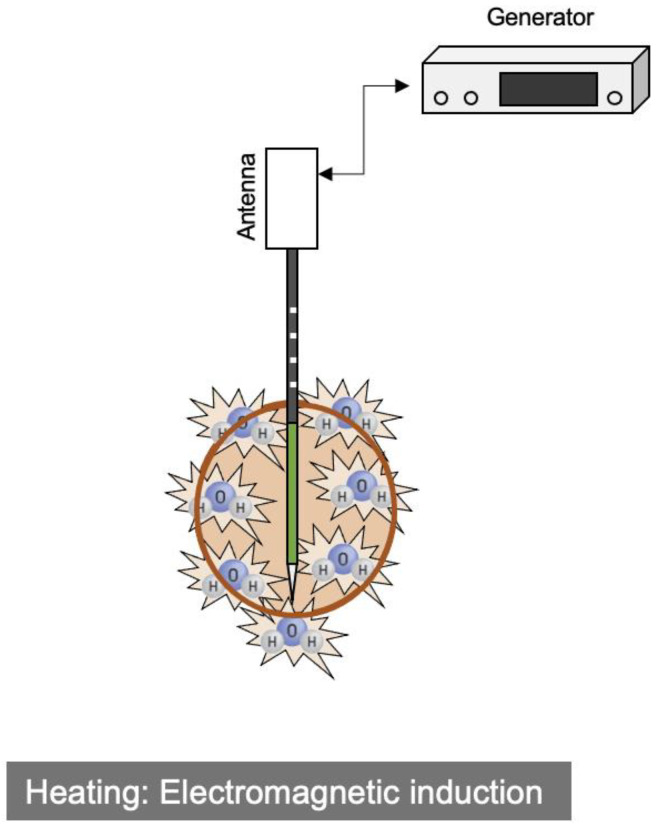
Schematic diagram of the MWA system and its mechanism of heat generation.

**Figure 3 cancers-15-04763-f003:**
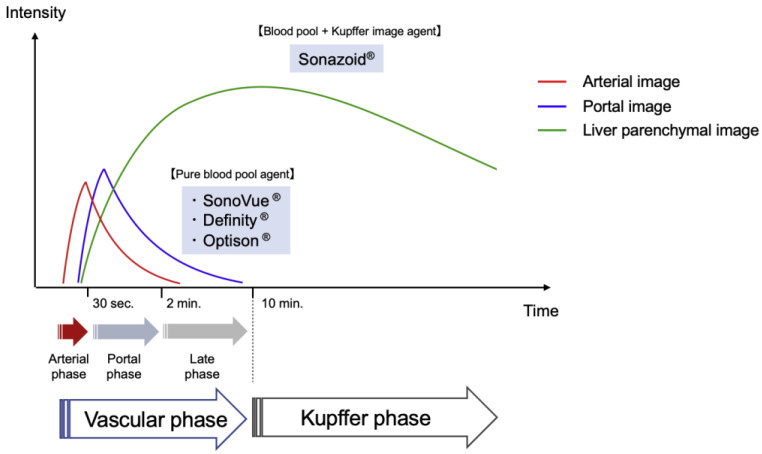
Pharmacokinetic behaviors of US contrast agents. The vascular phase shows tumor vascularity. The artery- and portal-dominant time zones in the vascular phase are referred to as the arterial and portal phases, respectively, while the Kupffer phase (from 10 min after the injection of Sonazoid^®^) shows hepatic parenchymal findings.

**Figure 4 cancers-15-04763-f004:**
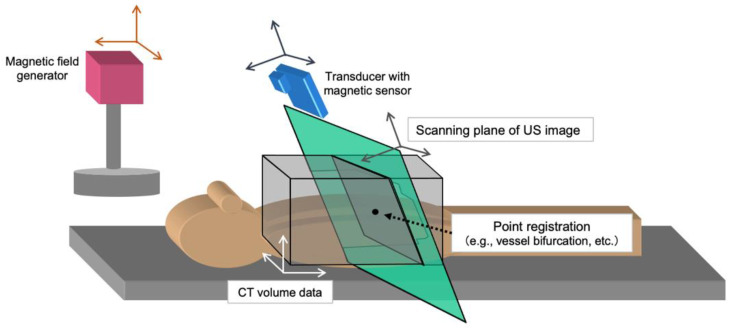
Schematic diagram of the image fusion system. The four coordinate systems of the CT volume (white), magnetic generator (orange), magnetic sensor (blue), and ultrasonic probe (green) are shown. Image fusion between US and MPR of CT images require the transformation matrixes in turn. Finally, synchronous imaging can be demonstrated for each movement of the US transducer. Note, CT: computed tomography, US: ultrasound, MPR: multiplanar reconstruction.

**Table 1 cancers-15-04763-t001:** Characteristics of RFA and MWA.

	RFA	MWA
Heat generation	Joule effect	Induction heating
Energy	Alternating current (450 kHz)	Electromagnetic waves (2.45 GHz)
Needle gauge	17 G	13 G (Emprint^®^); 14 G, 17 G, 18 G (Mimapro^®^)
Output voltage, W	~200 W	~100 W
Temperature, °C	~100 °C	~150 °C
Ablation zone	Oval	(Oval~) Sphere
Heat-sink effect	Strong	Weak
Grounding pads	Necessary	Unnecessary
Parameters on ablation	Energization time, voltage, tissue impedance	Energization time, voltage

## Data Availability

Not applicable.

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
