# Peer review of "Tips for Preparing and Practicing Thermal Ablation Therapy of Hepatocellular Carcinoma"

_cancers, 2023, doi:10.3390/cancers15194763_

Round 1
Reviewer 1 Report
1. The title includes the practice of thermal ablation therapy for hepatocellular carcinoma, but the article does not specifically discuss the entire complete thermal ablation process including image guidance.
2. Some relevant reviews on the progress of thermal ablation technology can be appropriately added to the introduction.
3. In Section 2.2, the existing microwave ablation instruments can be mentioned as in Section 2.1.
4. The part of proper methodology is not reflected in Section 3. There is also no distinction between the clinical indications for the use of the two ablation modalities (MWA and RFA).
5. In line 317 of the seventh section of the treatment response assessment, the treatment effect was evaluated by comparing the axial images before and after ablation, which were images of how long after ablation, and whether tissue contraction was considered. In clinical practice, whether follow-up observation is necessary or not, and how long the ablation has not recurred is considered successful.
6. The success rate of thermal ablation therapy for hepatocellular carcinoma was not mentioned in the entire article, and some additional points can be added appropriately to make the research more meaningful.
Author Response
Response to Reviewer #1
We wish to express our appreciation to the reviewer for providing insightful comments on our paper. The comments have helped us significantly improve the paper.
- The title includes the practice of thermal ablation therapy for hepatocellular carcinoma, but the article does not specifically discuss the entire complete thermal ablation process including image guidance.
We added some comments on a complete ablation process at the beginning of the section 6. [Page 8, line 271-277]
- Some relevant reviews on the progress of thermal ablation technology can be appropriately added to the introduction.
We added some comments on the treatment transition from PEI to thermal ablation in the introduction. [Page 2, line 42-50]
- In Section 2.2, the existing microwave ablation instruments can be mentioned as in Section 2.1.
We added some comments on antenna probes of Emprint and Mimapro in the subsection 2.2. [Page 4, line 111-113]
- The part of proper methodology is not reflected in Section 3. There is also no distinction between the clinical indications for the use of the two ablation modalities (MWA and RFA).
We have already described at the end of section 2 as below, “In clinical settings, MWA and RFA are considered to have equivalent efficacy, complication rates, local recurrence rates, and survival rates. Moreover, according to international HCC clinical practice guidelines, we could not find any difference in treatment indication between RFA and MWA. Therefore, we added a brief comment in the section 3. [Page 5, line 139-142]
- In line 317 of the seventh section of the treatment responseassessment, the treatment effect was evaluated by comparing the axial images before and after ablation, which were images of how long after ablation, and whether tissue contraction was considered. In clinical practice, whether follow-up observation is necessary or not, and how long the ablation has not recurred is considered successful.
We revised the sentences about monitoring of ablation using US fusion imaging as below, US–US fusion imaging allows side-by-side comparison of the ablative margin during the ablation, and this feedback helps operators to recognize residual tumors. Moreover, US–US overlay fusion can visualize the ablative margin immediately after ablation because the tumor image is projected onto the ablative hyperechoic zone. [Page 10, line 354-357]
Of course, tissue contraction causes by thermal ablation. However, during the ablation procedure, we did not consider the degree of tissue contraction because an error in measurement should be minimum in a human liver from our experience.
We added comments on a follow-up schedule. [Page 10, line 372-375]
- The success rate of thermal ablation therapy for hepatocellular carcinoma was not mentioned in the entire article, and some additional points can be added appropriately to make the research more meaningful.
We modified section 8 “Adverse events” to “Clinical outcomes and adverse events” and added some comments on the success rates of thermal ablation therapies. [Page 11, line 394-400]
Thank you again for your careful review of our manuscript. We look forward to receiving your further response.
Reviewer 2 Report
Very interesting and well written review on the technical aspects of thermal ablation of HCC.
I recommend to add more comments on the rationale underlying these treatments, with a focus on the potential immunomodulatory effects of ablative treatments.
The quality of the figures should be improved....maybe increasing the size would help
THe authors should add some tables for example summarizing the main comparative trials between RFA and MWA
The authors should add some comments on the clinical implications of thermal ablation therapies, for example on their role in improving post-recurrence survival after previous therapies (cite the paper PMID: 25085684)
The authors should cite the potential adjuvant therapies after RFA or MWA in HCC patients (cite the series PMID: 25974743)
Author Response
Response to Reviewer #2
We wish to express our appreciation to the reviewer for providing insightful comments on our paper. The comments have helped us significantly improve the paper.
- Very interesting and well written review on the technical aspects of thermal ablation of HCC. I recommend to add more comments on the rationale underlying these treatments, with a focus on the potential immunomodulatory effects of ablative treatments.
We added some comments by the new subsection, “2.3. Potential immunomodulatory effects”. [Page 4, line 124-129]
- The quality of the figures should be improved....maybe increasing the size would help.
The sizes of all figures were getting bigger and higher quality.
- The authors should add some tables for example summarizing the main comparative trials between RFA and MWA.
I added a table showing some characteristics between RFA and MWA. [Page 4-5]
- The authors should add some comments on the clinical implications of thermal ablation therapies, for example on their role in improving post-recurrence survival after previous therapies (cite the paper PMID: 25085684) The authors should cite the potential adjuvant therapies after RFA or MWA in HCC patients (cite the series PMID: 25974743)
We added some comments by the new section, “9. Clinical implications of thermal ablation therapies”. [Page 11, line 409-Page 12, line 435]
Thank you again for your careful review of our manuscript. We look forward to receiving your further response.
Round 2
Reviewer 2 Report
The revised version is OK. Thank you!